# A Systematic Review of the Association Between Pain and Instrumental Activities of Daily Living Disability in Community-Dwelling Older Adults

**DOI:** 10.3390/geriatrics10050113

**Published:** 2025-08-23

**Authors:** Yukiko Mizutani, Shigekazu Ukawa

**Affiliations:** 1Osaka Metropolitan University Graduate School of Human Life and Ecology, 1-1-138 Sugimoto, Sumiyoshi-ku, Osaka 558-8585, Japan; 2Kuwana City, Government, 2-37 Chochos, Kuwana 511-8601, Japan

**Keywords:** pain, instrumental activity of daily living, aged, systematic review

## Abstract

Introduction: Pain is highly prevalent among community-dwelling older adults and can undermine their ability to perform Instrumental Activities of Daily Living (IADL), which are essential for independent living. This systematic review aimed to summarize existing research to clarify the relationship between pain and IADL disability in community-dwelling older adults. Methods: We conducted a search of PubMed on 27 July 2025. Eligible studies met the following criteria: (1) assessed the association between pain and IADL disability; (2) included community-dwelling older adults aged 60 and older; and (3) were published in English. Results: Of the 400 records screened, 29 studies met the inclusion criteria. Of these, 23 studies (18 cross-sectional and 5 cohort studies) reported a significant association between pain and IADL disability, while 6 cross-sectional studies did not. Pain was assessed using diverse instruments across varying recall periods and thresholds, and IADL disability was measured using multiple scales. Such methodological heterogeneity precluded quantitative synthesis. Conclusions: In community-dwelling older adults, pain consistently predicts IADL disability across designs and settings. However, the lack of standardized, multidimensional measures and incomplete adjustment for treatment, multimorbidity, and polypharmacy limits precise effect estimation. Future research should adopt harmonized assessment tools, control comprehensively for relevant confounders, and perform meta-analyses where data permit to clarify pain’s true impact on functional independence.

## 1. Introduction

Pain is highly prevalent among community-dwelling older adults. A global systematic review found the prevalence to be between 25 and 76% across community-dwelling older population [1]. Estimates in the United States range from 27.6 to 33.6% among noninstitutionalized older adults [2], and up to half report pain in at least one body region within the past month [3]. In the United Kingdom, prevalence ranges from 35.0 to 51.3% (pooled estimate 43.5%; 95% CI: 38.4–48.6%) and reaches 62% among those aged 75 and older [4]. Across Europe, prevalence among adults aged 50 and older varies between 30% and 60%, with annual increases of 2.2 to 5.8% from 2004 to 2015 [5]. In China, the prevalence of pain among people aged 45 years or older was 60% [6]. In Japan, 39.0% of independent older individuals report pain (men 36.3%, women 41.8%), with prevalence increasing with age [7].

Instrumental Activities of Daily Living (IADL) are a frequently used measure to assess the ability of individuals, including older adults living independently, to perform the activities of daily living necessary for independent living [8,9]. In the United States, 37.1% of adults aged 60 years or older experienced difficulty with at least one IADL [10]. In the United Kingdom, limitations in at least one IADL were reported by 18.3% of men and 24.5% of women [11]. Across Europe, 23.8% of individuals aged 65 years or older had 1 or more IADL impairments, with a higher prevalence among women (27.1%) compared to men (17.6%), and rates rising to 51.5% among those aged 85 years and above [12]. The prevalence was 32% in China among adults aged 65 or older [13], and 15.9% in Japan among those aged 75 or older, with 10.7% developing functional disability over a 24-month period [14]. A decline in IADLs is associated with various adverse health outcomes, including dementia [15], mild cognitive impairment [16], frailty [17], depression [18], and mortality [19]. Experiencing pain may negatively affect IADL function through limited physical activity [20,21], impaired mental health [22,23,24,25], impaired cognitive function [26,27,28], and decreased social participation [29,30]. Pain management and maintenance of IADL function are crucial for promoting healthy aging and reducing the burden on healthcare systems.

Persistent or multisite pain sharply increases the risk of new IADL limitations in community-dwelling older adults, with cohort studies reporting two- to three-fold higher odds of decline within 2–5 years [31]. Several mechanisms plausibly link pain to IADL disability. Persistent pain is associated with attentional and short-term memory deficits [26,32], making complex instrumental tasks especially vulnerable to decline [27,28]. Pain also heightens depressive symptoms [23,33] and leads to reduced physical activity and social participation [20,21,29], pathways that further erode functional independence [30]. Clarifying the relationship between specific pain characteristics and functional independence remains essential; the present review addresses this gap.

Although multiple observational studies have examined the association between pain and IADL disability among older adults, their findings have been fragmented. To date, no comprehensive synthesis of the available evidence has been conducted. Therefore, this systematic review aimed to summarize existing research to clarify the relationship between pain and IADL disability in community-dwelling older adults.

## 2. Materials and Methods

This systematic review was conducted in accordance with the Preferred Reporting Items for Systematic Reviews and Meta Analyses (PRISMA) guidelines and was registered on the PROSPERO platform under the code CRD420251072156.

### 2.1. Search Strategy

A systematic literature search was conducted in the PubMed electronic database on 27 July 2025. The search included studies published up to that date. The following Medical Subject Headings (MeSH) and free-text terms were used in combination: “pain” [MeSH Terms] AND (“Activities of Daily Living”[MeSH Terms] OR “instrumental activities of daily living”[All Fields] OR “IADL” [All Fields]).

### 2.2. Eligibility Criteria

Studies were included if they met the following criteria: (1) assessed the association between pain and disability in instrumental activities of daily living (IADL); (2) included community-dwelling older adults aged 60 and older [34] who were able to provide self-reported data, which effectively excluded individuals who were bedridden, in terminal stages, or severely ill, and (3) were published in English.

### 2.3. Study Selection

The studies identified through the systematic review were independently screened by two researchers (YM and SU) based on their titles, abstracts, and full texts. Any discrepancies were resolved through discussion.

### 2.4. Data Extraction

Two reviewers (YM and SU) independently extracted data from the included studies. The following information was collected: first author, publication year, country, study design, sample size, mean age of participants, pain assessment (e.g., presence, location, severity), IADL assessment method, covariates adjusted for, statistical methods used, and key findings regarding the association between pain and IADL disability. Any disagreements were resolved through discussion.

### 2.5. Data Synthesis 

Because pain definitions, IADL instruments, and analytical approaches varied greatly, we conducted a narrative synthesis. Each study was classified as showing a significant association, a non-significant association, or inconsistent findings, using *p* < 0.05 as the threshold for statistical significance. The findings were then summarized by study design and by pain dimension to provide a narrative subgroup synthesis.

### 2.6. Quality Assessment

The methodological quality of the included studies was not formally assessed, as the purpose of this review was to provide a descriptive summary of the existing literature rather than to perform a meta-analysis.

## 3. Results

### 3.1. Identification of Studies

A total of 400 studies were identified through the PubMed database. After screening titles and abstracts, 334 articles were excluded. The full texts of 66 articles were reviewed, and 29 studies met the inclusion criteria [23,24,25,28,31,35,36,37,38,39,40,41,42,43,44,45,46,47,48,49,50,51,52,53,54,55,56,57,58]. The main reasons for the exclusion of full-text articles were as follows: the study did not assess the association between pain and IADL disability, the participants were under 60 years of age, the participants were not community-dwelling older adults, or the study population consisted of patients. The numbers for each exclusion reason are shown in Figure 1.

### 3.2. Study Characteristics

Table 1, Table 2 and Table 3 summarize the characteristics of the included studies. 3 studies were published each in 2019 [31,46,47], 2014 [23,51,52], and 2010 [24,53,54]; 2 studies were published each in 2025 [35,36], 2024 [37,38], 2023 [39,40], 2021 [42,43], 2020 [44,45], and 2018 [48,49]; and 1 study was published each in 2022 [41], 2017 [28], 2016 [50], 2009 [55], 2006 [56], 2004 [25], 2000 [57], and 1992 [58]. 10 studies were conducted in the United States [23,31,37,38,48,51,52,53,55,57], 4 studies were conducted in China [25,35,41,45], 3 studies were conducted in Poland [36,46,49]; 2 studies each were conducted in Nigeria [40,56] and Canada [24,54]; and 1 study each was conducted in India [39], Australia [42], Sweden [43], Saudi Arabia [44], Spain [47], Ireland [28], Singapore [50], and France [58]. The sample sizes ranged from 171 [44] to 31,464 [39] (Table 1).

### 3.3. Design

The included studies comprised both cohort [31,42,48,52,53] and cross-sectional designs [23,24,25,28,35,36,37,38,39,40,41,43,44,45,46,47,49,50,51,54,55,56,57,58] (Table 1).

### 3.4. Pain Assessment

#### 3.4.1. Pain Presence Definition

The included studies used a variety of definitions and instruments to assess pain presence, such as pain scale [23,24,36,42,44,45,46,49,52,54,55,57], self-reported discomfort [35], bother [38] or trouble [25,28,39], pain restricting activity [48], pain present on most days [53], presence of pain without additional criteria [37,40,41,43,47,50,56,58], and pain quality [31] or pain location [51] (Table 2).

#### 3.4.2. Pain Location

Pain location did not specify in some studies [28,31,36,37,39,40,45,46,47,49,57], while others reported specific location such as back, waist, buttocks, hips, thigh, legs, knees, feet, ankle, toes, shoulders, elbow, arms, hands, wrists, fingers, neck, head, generalized, face or tooth or jaw, stomach, abdominal, chest, body, spine, lower extremities, joints, muscles, and bones [23,24,25,35,38,41,42,43,44,48,50,51,52,53,54,55,56,58] (Table 2).

#### 3.4.3. Pain Severity

Pain severity was assessed using several instruments, including the Visual Analog Scale [46,49,51], Verbal Descriptor Scale [23,24,54], Euroqol 5D quality of life assessment questionnaire pain scale [36], Short Form Health survey-12 [42], six-point Likert scale [44], Numeric Rating Scale [45], Brief Pain Inventory [52], Short Form Health survey-36 [55], and McGill Pain Questionnaire [57] (Table 2).

#### 3.4.4. Pain Frequency/Quality

Pain frequency was categorized into four levels in one study [39], and pain quality was assessed using the MOBILIZE Boston Study pain quality instrument [31] in another study (Table 2).

### 3.5. IADL Assessment

IADL disability was assessed using a variety of scales, including the Lawton IADL scale [25,28,31,35,36,40,41,43,45,46,47,49,50,51,57,58], the Older Americans Resources and Services scale [24,53,54], the Rosow-Breslau IADL scale [42], the Nagi Physical Performance Scale and the Health Assessment Questionnaire [56], as well as a set of individual activities [23,37,38,39,44,48,52,55]. Response categories and scoring methods also varied, with some studies using dichotomized [24,28,35,38,39,40,41,42,45,46,47,48,49,50,52,53,56,58], or trichotomized [31,57] responses, and other evaluating total scores [23,25,36,37,43,44,51,54,55] (Table 3).

### 3.6. Confounders

A wide range of variables were adjusted for in the multivariable analysis: age [24,25,28,31,35,36,39,42,45,46,47,48,49,50,52,53,54,55,56,57,58], sex [31,35,36,39,45,47,48,50,52,53,54,55,56,57,58], education [24,28,31,35,36,39,45,47,48,49,50,52,53,54,55,58], marital status [28,36,39,45,50], living arrangements [28,39,48,50], work status [28,39,50], income [50], ethnicity or race [24,31,48,50,52], religion [39], place of residence [39,58], region [39], social group [39], and wealth quintiles [39]. Health-related factors included self-rated health [24,28,35,39,57], number of comorbidities [24,35,42,46,48,49,54,55,57], presence of chronic diseases [28,39,50,52], diabetes [31,50], hypertension [50], hyperlipidemia [50], lung disease [31], heart disease [31], vascular diseases [53], vascular risk factors [53], visual and hearing impairments [28,50,58], body mass index [28,31,42,45,50,52,53,55], depression [28,50,54,58], depressive symptoms [24,25,38,39,42,48,53,55], cognition [28,35,39,48,50,52,53,54,55], Mini-Mental State Examination score [31], self-rated memory [28], worry levels [28], social support [50], social contacts [46], social connectedness [28], loneliness [28], good relations with relatives [46], presence of barriers and obstacles [46], adaptation of the home environment [49], physical activity [28,31,35,39,42,46,49,52,53], exercise [40,45], restriction of habitual activity [47], bedridden status [47], frailty [48,55], history of falls [28,35,49,50], smoking [28,42], medication use (number of drugs or prescription medications) [28,36,42,52], assistive device use [49], time spent sitting [28], life satisfaction [35], and quality of life [28,49] (Table 3).

### 3.7. Statistical Analysis

15 studies were analyzed using the logistic regression [24,25,28,35,38,39,42,45,46,47,49,50,56,57,58]. 2 studies each were analyzed using the Poisson regression [31,52] and the Cox proportional hazards model [48,53]. 1 study each was analyzed using the Wald test [23], the Analysis of Variance [36], the Hierarchical regression [37], Fisher’s exact test [40], the chi-square test [41], the Mann–Whitney U test [43], the Kruskal–Wallis test [44], the *t*-test [51], the linear regression [54] and the negative binomial regression [55] (Table 3).

### 3.8. Synthesis of Results

A total of 29 studies were included in this review. Of these, 23 studies reported a significant association between pain and IADL disability; they consisted of 18 cross-sectional [23,24,35,36,37,38,39,41,44,45,46,47,49,50,51,54,55,56] and 5 cohort [31,42,48,52,53] studies, whereas the remaining 6 studies did not [25,28,40,43,57,58].

6 cross-sectional studies reported no significant relationship between pain presence and IADL disability [25,28,40,43,57,58]. In these studies, 3 defined pain presence simply as the occurrence of pain [40,43,58], 2 defined it as being troubled by pain [25,28], and 1 assessed pain presence using the McGill Pain Questionnaire pain scale [57]. In the studies reporting a significant association, pain presence was defined by a pain scale in 50% of the cross-sectional studies [23,24,36,44,45,46,49,54,55] and in 60% of the cohort studies [31,42,52]. Taken together, these observations provide a narrative summary, indicating that the association between pain and IADL disability is more consistently observed in cohort studies and in studies that defined pain with validated scales.

#### 3.8.1. Association Between Pain Location and IADL Disability

5 studies that targeted 5 to 15 pain areas either classified participants into groups based on the number of pain location [35,38,41,52] or calculated the risk of IADL disability associated with each additional site [53]. 5 studies measured pain location by simply selecting 3 to 13 locations [25,42,50,56], or by using a pain map [23], and assessed IADL disability based on the presence of pain without considering the number of pain location. 5 studies measured body pain [24,44,54,55] or joint pain [58] without specifying exact locations; among these, 4 studies [24,44,54,55] evaluated pain severity using assessment scales. Overall, these 14 studies indicated that pain was significantly associated with IADL disability, except for 2 studies [25,58]. 2 studies measured pain at a specific location on the back [43,48]. Among these studies, 1 reported a significant association [48] while 1 reported no significant association [43]. 11 studies [28,31,36,37,39,40,45,46,47,49,57] that did not specify the pain location generally indicated a significant association between pain and IADL disability, except for 3 studies [28,40,57]. 1 study compared IADL scores between groups with pain at two locations (spinal pain versus those with lower extremity pain) [51] and found no significant difference between the groups.

#### 3.8.2. Association Between Pain Severity and IADL Disability

9 studies did not use pain severity scales. Of these, 6 found that pain was significantly associated with IADL impairment [37,41,47,50,53,56], while 3 did not [40,43,58]. 5 studies evaluated pain severity by asking how discomfort [35], bothered [38] or troubled [25,28,39] participants were by pain; in 3 studies [35,38,39], a significant association was reported, while 2 studies [25,28] no significant association. 1 study defined pain severity as the restriction of activity due to pain [48], and found a significant association. 13 studies used pain severity scales [23,24,36,42,44,45,46,49,51,52,54,55,57]. Most of these studies reported a significant association between pain and IADL disability, except for 1 study [57]. 1 study found that a 1-point increase on the Visual Analog Scale was associated with 1.27 times higher odds of disability (95% CI: 1.22–1.33) [46], while another study reported that a 1-point increase was associated with a1.21 times higher odds of disability (95% CI: 1.06–1.36) [49].

#### 3.8.3. Association Between Pain Frequency/Quality and IADL Disability

1 study categorized pain frequency into four levels and found that, compared with individuals without pain, the odds of IADL disability were 1.12 (95% CI: 1.02–1.23) for rare pain, 1.49 (95% CI: 1.38–1.61) for occasional pain, and 1.67 (95% CI: 1.53–1.82) for frequent pain [39]. 1 study measured pain quality by the MOBILIZE Boston Study and found that, at 18 months follow-up, individuals with 2 persistent pain qualities had a relative risk of 2.59 (95% CI: 1.10–6.09) and those with 3 had a relative risk of 2.69 (95% CI: 1.34–7.79) compared with 1 persistent pain quality [31].

## 4. Discussion

This systematic review synthesized findings from 29 studies investigating the association between pain and IADL disability in community-dwelling older adults. The majority of these studies (23 out of 29) reported a significant relationship, highlighting the considerable impact of pain on functional independence.

Several potential mechanisms may explain the observed association between pain and IADL dysfunction. First, persistent pain has been associated with memory decline [26], which in turn is linked to IADL disability [28]. Key features of cognitive dysfunction related to chronic pain include reduced attentional capacity and impaired short-term memory [32]. Pain may compete for limited cognitive resources and divert attention from cognitive tasks, especially in cases of severe pain or frequent rumination, leading to memory impairment due to incomplete encoding [26,32,59]. Given that IADL tasks require more complex neuropsychological processing than basic ADLs, these cognitive deficits make IADLs particularly vulnerable to decline [27]. Second, older adults with pain are more likely to experience depressive symptoms [23], which in turn are associated with an increased risk of IADL limitations [33]. Because IADL tasks place greater cognitive demands than basic ADLs or mobility tasks, this may partly explain the association between depressive symptoms and IADL limitations. Structural equation modeling suggests that depressive symptoms may impair IADL performance indirectly by reducing cognitive function [33]. Third, chronic pain is associated with reduced physical activity [20], which may lead to IADL impairment through mechanisms such as increased fatigue, decreased motivation, and cognitive decline [21,28]. It is also linked to social frailty, as older adults with pain are less likely to engage in social activities like going out or visiting friends [29]. In contrast, greater social participation is associated with lower IADL disability by encouraging daily instrumental activities, enhancing access to resources and health-related information, and reducing psychological stress through emotional support [30,60].

Pain measurement in the included studies had several systematic and methodological limitations. All 29 studies relied solely on self-reported scales without incorporating objective measures of pain or disability, which may have introduced recall and reporting bias and led to underestimation of the true pain burden. Because pain is inherently subjective and no standardized instrument exists, heterogeneity may have arisen from variation in question wording, the use of unvalidated tools with inconsistent recall periods (e.g., past 3 months vs. 4 weeks) [42]. Furthermore, simplifying explanatory variables into binary categories and using ordinal verbal descriptor scales [23] limit interpretability by obscuring differences between levels of severity. Limitations in existing studies include the inability to determine the temporal relationship between persistent pain and functional decline [31], missing information on pain location [24], and the absence of physiological indicators in tools like the Short Form Health survey-36 [55]. Self-reported pain is particularly prone to underreporting among older adults with cognitive impairment, who may struggle to perceive or articulate pain [61]. Non-standardized definitions further obscure prevalence: for example, merely altering the knee pain question wording changed prevalence from 19% to 28% in the same population [62], and extending the recall period from 3 to 6 months increased prevalence estimates by 30% [63]. If pain is under-recognized or inconsistently defined, its true impact on IADL function may be underestimated. Therefore, addressing this issue requires the use of standardized, validated, and objective tools capable of capturing the multidimensional nature of pain in older adults.

Although most studies adjusted for demographic and health covariates, only one [52] accounted for daily analgesic use, and none considered non-pharmacological treatments such as physical or occupational therapy. Similarly, only two studies adjusted for the number of prescription medications [42] or polypharmacy (defined as ≥5 medications) [28]. Regarding multimorbidity, 9 studies used a comorbidity count [24,35,42,46,48,49,54,55,57]. This limited adjustment likely results in residual confounding by overall health burden, as older adults with complex disease profiles are predisposed to both pain and IADL disability. Inadequate adjustment for factors such as pain treatment, polypharmacy, and disease clustering may bias current estimates of pain’s impact on IADL disability. The association could be overstated if untreated pain is unaccounted for, or understated if comorbidities are over-adjusted. Future research should gather detailed data on pain management, use comprehensive multimorbidity indices, and consider medication burden to better isolate pain’s independent contribution to IADL decline.

This review has several strengths. First, the review examined multiple dimensions of pain, including location, frequency, severity, quality, and presence, in relation to IADL disability. This approach provided a comprehensive understanding of how various aspects of pain may influence functional independence. Second, by incorporating both cross-sectional and cohort studies, the review captured concurrent associations as well as temporal relationships between pain and IADL decline. However, this review also has certain limitations. First, the search for relevant studies was limited to those published in English, which may have resulted in the exclusion of relevant studies published in other languages. Second, we did not conduct a formal risk-of-bias assessment or perform a quantitative meta-analysis. A meta-analysis was not feasible due to considerable heterogeneity among the 29 included studies. Pain was assessed using various instruments, with differing recall periods and thresholds. IADL disability was measured using a range of tools and categorized as dichotomous, trichotomous, or continuous outcomes. Effect estimates were reported in multiple formats, including odds ratios, relative risks, hazard ratios, and regression coefficients, and were often presented without sufficient variance measures. Additionally, the timing of pain and IADL assessments varied widely, and covariate adjustment differed considerably between studies. These methodological inconsistencies made it inappropriate to pool the data. Future research should apply standardized measurement tools, report effect estimates with accompanying variances and conduct structured quality appraisals. Third, this review was based on a literature search conducted using only PubMed. Although efforts were made to identify relevant studies through broad search terms and reference list screening, the possibility remains that some relevant studies indexed in other databases were not captured. Future reviews may benefit from incorporating additional databases such as Embase, CINAHL, PEDro, or Web of Science to enhance comprehensiveness and reduce the risk of publication bias. Fourth, the search for relevant studies was limited to those published in English, based on the recommendation of the Cochrane Handbook for Systematic Reviews of Interventions [64], which notes that excluding in other languages usually does not affect review conclusions. However, we acknowledge that this criterion may still have introduced language bias and could have resulted in the exclusion of pertinent non-English studies. Fifth, because IADLs are more complex than basic ADLs and require a certain level of independence, our findings may not be generalizable to older adults who are bedridden or have severe illnesses. Sixth, we did not conduct a formal methodological quality assessment of the included studies due to the diversity in study designs and outcomes. This may have introduced additional bias into our findings. Seventh, we were unable to conduct a quantitative subgroup meta-analysis owing to substantial heterogeneity in outcome scales and reported effect estimates. Eighth, our findings may not be generalizable to older adults who are bedridden, at the end of life, or severely ill, because the requirement for independent self-report effectively excluded these groups.

## 5. Conclusions

Pain is a consistent predictor of IADL disability in community-dwelling older adults. Despite variability in populations, pain definitions, and functional assessments, both cross-sectional and longitudinal data support this association. However, heterogeneity in measurement tools and limited adjustment for treatment, multimorbidity, and polypharmacy precluded meta-analysis and precise effect estimation. Future research should use standardized, multidimensional measures, account for relevant confounders, and conduct meta-analyses where possible to better determine pain’s impact on independent functioning in later life.

## Figures and Tables

**Figure 1 geriatrics-10-00113-f001:**
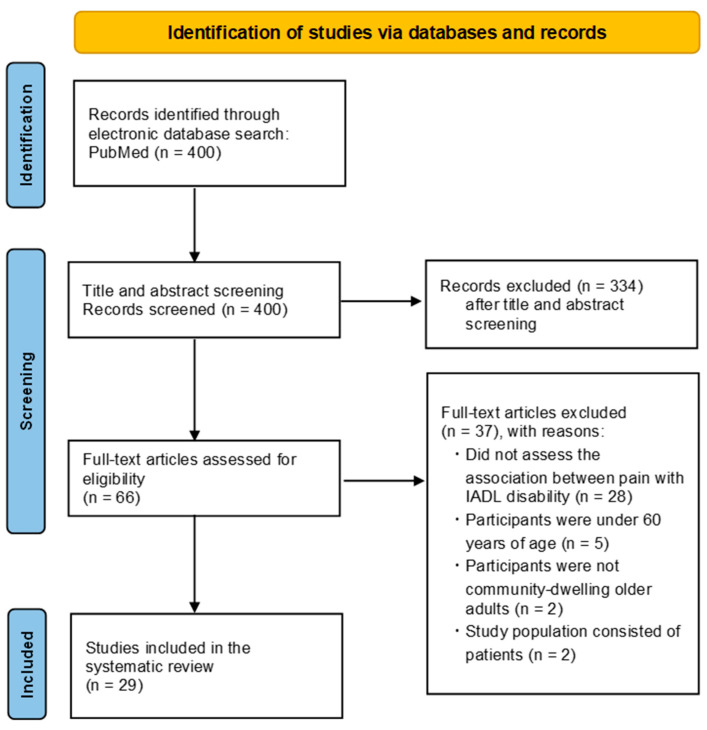
Flow Diagram of the Study Selection Process.

**Table 1 geriatrics-10-00113-t001:** Characteristics of included studies.

First Author	Year	Location	Design	Sample Size	Female (%)	Mean Age (SD), Range
Chu J. [35]	2025	China	Cross-sectional	8102	49,7	68.0, 64–73
Balicki P. [36]	2025	Poland	Cross-sectional	2992	66.6	-, 60–106
Ord AS. [37]	2024	USA	Cross-sectional	452	56.0	73.5, 60–89
Liu P. [38]	2024	USA	Cross-sectional	5557	59.4	-, ≥65
Muhammad T. [39]	2023	India	Cross-sectional	31,464	52.5	-, ≥60
Olawumi AL. [40]	2023	Nigeria	Cross-sectional	312	59.6	67.67 (7.69), 60–74
Lu Z. [41]	2022	China	Cross-sectional	7619	51.2	68.2 (6.3), ≥60
Scott D. [42]	2021	Australia	Cohort	1452	0	76.5 (5.2), ≥70
Svensson HK. [43]	2021	Sweden	Cross-sectional	446	100	70.0 (0), 70
Al-Qahtani AM. [44]	2020	Saudi Arabia	Cross-sectional	171	0	70.6, 60–102
Peng X. [45]	2020	China	Cross-sectional	1321	63.8	71.6 (9.3), ≥60
Ćwirlej-Sozańska A. [46]	2019	Poland	Cross-sectional	2207	60.0	72.1 (7.8), ≥60
Carmona-Torres JM. [47]	2019	Spain	Cross-sectional	25,465	60.6	75.9 (7.4), ≥65
Thakral M. [31]	2019	USA	Cohort	398	-	-, ≥70
Makris UE. [48]	2018	USA	Cohort	754	64.6	78.4(5.3), ≥70
Ćwirlej-Sozańska Ab. [49]	2018	Poland	Cross-sectional	426	59.9	75.6 (2.9), 71–80
Connolly D. [28]	2017	Ireland	Cross-sectional	3499	52.5	-, ≥65
Liang En W. [50]	2016	Singapore	Cross-sectional	559	55.3	-, ≥60
Shega JW. [23]	2014	USA	Cross-sectional	2430	52.3	-, ≥62
Yağci N. [51]	2014	USA	Cross-sectional	258	49.2	72.0 (5.9), 65–100
Eggermont LH. [52]	2014	USA	Cohort	634	64	78.0 (5.0), 64–97
Shega JW. [24]	2010	Canada	Cross-sectional	5549	-	-, ≥65
Buchman AS. [53]	2010	USA	Cohort	898	74.3	79.7, ≥65
Shega JW. [54]	2010	Canada	Cross-sectional	5086	-	-, ≥65
Weaver GD. [55]	2009	USA	Cross-sectional	744	63	82.0 (4.4), 74–100
Gureje O. [56]	2006	Nigeria	Cross- sectional	2152	47.5	75.0 (9.2), ≥65
Miu DK. [25]	2004	China	Cross-sectional	749	51.4	75.2 (6.6), ≥65
Mossey JM. [57]	2000	USA	Cross-sectional	228	80.7	-, ≥60
Barberger-Gateau P. [58]	1992	France	Cross-sectional	2792	60	-, ≥65

SD = Standard Deviation. Note: “-” indicates data not reported in the original publication.

**Table 2 geriatrics-10-00113-t002:** Overview of pain assessment.

First Author	Pain Presence Definition	Pain Location	Pain Severity	Pain Frequency/Quality
Chu J. [35]	Experiencing any pain-related discomfort	Head, neck, shoulder, arm, wrist, fingers, chest, stomach, back, waist, buttocks, leg, knees, ankle, toes, and other specified areas	Without vs. with Without pain vs. pain in the head, upper limb, torso, or lower limb Without pain vs. single-site pain vs. multisite pain	-
Balicki P. [36]	EQ-5D pain scale: moderate or extreme problems	-	Without vs. with	-
Ord AS. [37]	Having chronic pain complaints	-	Without vs. with	-
Liu P. [38]	Having bothered pain	Back, hips, knees, feet, hands, wrists, shoulders, neck, arms, legs	Multisite pain (≥2 sites) vs. no multisite pain (≤1 site)	-
Muhammad T. [39]	Being often troubled by pain	-	Without vs. with	Frequency: never, rarely (1–2 days per week), occasionally (3–4 days per week), frequently (≥5 days per week)
Olawumi AL. [40]	Having chronic pain	-	Without vs. with	-
Lu Z. [41]	Having body pain in any, or in back and/or shoulder, or in leg and/or knee pain	Head, back, shoulder, arm, stomach, legs, neck, knees, others	Without vs. with	-
Scott D. [42]	During the past 4 weeks, having pain interfered with normal work	Body: hands, wrists, elbows, shoulders, face, jaw, neck, hips, knees, ankles, feet, back, other	SF-12 intrusive pain: moderately, quite a bit, extremely	-
Svensson HK. [43]	Having long-term pain	Back	Without vs. with	-
Al-Qahtani AM. [44]	Self-rated pain ≥ 1 point	Body	Six-point Likert scale: none, very mild, mild, moderate, severe, very severe	-
Peng X. [45]	NRS ≥ 1 point	-	NRS: 0 (no pain) to 10 (worst pain imaginable)	-
Ćwirlej-Sozańska A. [46]	VAS for pain sensation (ICF b280) ≥1 point	-	VAS: 0 (no pain) to 10 (worst pain imaginable)	-
Carmona-Torres JM. [47]	Having pain in the past 4 weeks	-	Without vs. with	-
Thakral M. [31]	-	-	-	MBS pain quality instrument: 20 descriptors in 3 groups (cognitive/affective, sensory, neuropathic); same quality (1, 2, or 3 categories)
Makris UE. [48]	Restricting activity by pain	Back	Without vs. with	-
Ćwirlej-Sozańska Ab. [49]	VAS for having pain in the last 30 days ≥1 point	-	VAS: 0 (no pain) to 10 (the worst pain imaginable)	-
Connolly D. [28]	Being often troubled by pain	-	Without vs. with	-
Liang En W. [50]	Having chronic pain lasting ≥3 months	Generalized, headache, face or tooth or jaw, neck, shoulder or elbow, arm or hand, chest, back, abdominal, hip or thigh, knee or leg, ankle or foot	Without vs. with	-
Shega JW. [23]	Having moderate or greater pain in the past 4 weeks	45 locations (specified on human body diagram)	VDS: no pain, slight pain, mild pain, moderate pain, severe pain, extreme pain, the most intense pain imaginable	-
Yağci N. [51]	Having spinal pain vs. lower extremity pain	Spinal pain (Group I), lower extremity pain (Group II)	VAS: 0 (no pain) to 10 (the worst pain imaginable)	-
Eggermont LH. [52]	1.Having single-site or wider spread pain lasting 3 or more months in the previous year and still present in the previous month 2. BPI severity subscale score >0 point 3. BPI interference with daily activities subscale >0 point	1. Back, chest, shoulder, hand/wrist, hip, knee, foot 2 and 3. -	1. 14-item questionnaire: no pain, single-site pain, >1 site not meeting criteria for widespread pain, or widespread pain (back or nonanginal chest pain) 2. BPI pain severity: 0 (no pain) to 10 (severe or excruciating pain as bad as imaginable) 3. BPI interference: 0 (not interfere at all) to 10 (complete interference)	-
Shega JW. [24]	Having very mild or greater noncancer pain in the past 4 weeks	Body	VDS: none, very mild, moderate, severe, very severe	-
Buchman AS. [53]	Having joint pain on most days for at least one month during the prior year	Joints (back or neck, hands, hips, knees, feet)	Without vs. with	-
Shega JW. [54]	Having very mild or greater noncancer pain in the past 4 weeks	Body	VDS: none, very mild, moderate, severe, very severe	-
Weaver GD. [55]	Having very mild or greater bodily pain in the past 4 weeks	Body	SF-36: none, very mild, mild, moderate, severe, very severe	-
Gureje O. [56]	Having persistent pain with 6 months duration	Back or neck, chest, joint, frequent headaches, general category of persistent pain in any other body parts	Without vs. with	-
Miu DK. [25]	Being troubled by pain in the previous 2 weeks	Joints, muscles, bones	Without vs. with	-
Mossey JM. [57]	Bothering pain with activity limitations in the last two weeks	-	McGill Pain Questionnaire: no pain, pain without activity, pain with activity limitations	-
Barberger-Gateau P. [58]	Suffering from pain	Joints	Without vs. with	-

Abbreviations: EQ-5D = Euroqol 5D quality of life assessment questionnaire; VAS = Visual Analog Scale; ICF = International Classification of Functioning, Disability and Health; SF = Short Form Health survey; NRS = Numeric Rating Scale; VDS = Verbal Descriptor Scale; BPI = Brief Pain Inventory; MBS = MOBILIZE Boston Study; ISA = The Ibadan Study of Aging Note: “-” indicates data not reported in the original publication.

**Table 3 geriatrics-10-00113-t003:** Association between pain and disability in instrumental activities of daily living (IADL).

First Author	IADL Assessment	Confounders	Statistical Analysis	Findings
Chu J. [35]	Lawton’s index (no disability vs. disability)	Age, gender, education, depression, comorbidities, self-report health, life satisfaction, physical activity, falls, cognition	Logistic regression	Participants with pain had a higher risk of IADL disability (aOR: 1.91, 95% CI: 1.67–2.19). Participants with pain located in the head and neck (aOR: 1.22, 95% CI: 1.06–1.41), upper limb (aOR: 1.25, 95% CI: 1.08–1.45), torso (aOR: 1.27, 95% CI: 1.10–1.46), and lower limb (aOR: 1.38, 95% CI: 1.20–1.59) had a higher odds of IADL disability. Participants with multisite pain (aOR: 2.13, 95% CI: 1.85–2.45) and single site pain (aOR: 1.35, 95% CI: 1.12–1.64) had progressively higher odds of IADL disability.
Balicki P. [36]	Lawton’s index (total score ranging from 0 (dependent) to 8 (independent))	-	Analysis of Variance	Participants with pain had significantly lower scores than those without pain (*p* < 0.001).
Ord AS. [37]	A dichotomized IADL self-report questionnaire (total score of items requiring assistance)	Age, gender, education, marital status, number of medical conditions	Hierarchical regression	Participants with chronic pain had higher IADL disability scores (β: 0.86, 95% CI: 0.31–1.41).
Liu P. [38]	5-item scale (no disability vs. disability)	Social participation, depressive symptoms, anxiety symptoms	Logistic regression	Participants with multisite musculoskeletal pain had higher odds of IADL disability (aOR: 1.99, 95% CI: 1.69–2.34).
Muhammad T. [39]	7-item scale (no difficulty vs. 1 difficulty)	Age, sex, education, marital status, living arrangements, work status, physical activity, self-rated health, chronic diseases, depressive symptoms, cognitive impairment, wealth quintiles, religion, social group, place of residence, regions	Logistic regression	Participants with pain had higher odds of IADL disability (aOR: 1.43, 95% CI: 1.35–1.51). Participants with rare pain (aOR: 1.12, 95% CI: 1.02–1.23), occasional pain (aOR: 1.49, 95% CI: 1.38–1.61), and frequent pain (aOR: 1.67, 95% CI: 1.53–1.82) had progressively higher odds compared to those without pain.
Olawumi AL. [40]	Lawton’s index (independent vs. dependent)	-	Fisher’s Exact test	No significant association was found for chronic pain.
Lu Z. [41]	Lawton’s index (no disability vs. at least one disability)	-	Chi-square test	Participants with any pain, back and/or shoulder pain, and leg and/or knee pain had significantly more disability (*p* < 0.001).
Scott D. [42]	Rosow-Breslau scale (no disability vs. disability)	Age, BMI, current smoking status, physical activity, number of comorbidities, number of prescription medications, depression symptoms	Logistic regression	At 5-year follow-up, participants with persistent intrusive pain (aOR: 4.63, 95% CI: 2.22–9.65) and incident intrusive pain (aOR: 2.98, 95% CI: 1.81–4.90) had higher odds of IADL disability.
Svensson HK. [43]	Lawton’s index (total score ranging from 0 (dependent) to 8 (independent))	-	Mann–Whitney U test	Not significant association was found for long-term back pain.
Al-Qahtani AM. [44]	7-item score (ranging: 7 = no impairment to 21 = total impairment)	-	Kruskal–Wallis test	Participants with severe body pain showed significantly worse IADL scores (*p* < 0.001).
Peng X. [45]	Lawton’s index (no disability vs. any disability)	Age, gender, marital status, education, exercise, BMI	Logistic regression	Participants with some degree of pain had higher odds of IADL disability (aOR: 2.97, 95% CI: 2.31–3.83).
Ćwirlej-Sozańska A. [46]	Lawton’s index (no difficulties vs. ≥1 difficulty)	Age, number of chronic diseases, physical activity, presence of barriers and obstacles, social contacts, good relations with relatives	Logistic regression	Each 1-point increase on the VAS was associated with higher odds of disability (aOR: 1.27, 95% CI: 1.22–1.33).
Carmona-Torres JM. [47]	Lawton’s index (no difficulty vs. ≥1 difficulty)	Age, gender, educational level, restriction of habitual activity, bedridden status	Logistic regression	Participants with pain in the past 4 weeks had higher odds of disability (aOR:2.8, 95% CI: 2.53–3.09).
Thakral M. [31]	Lawton’s index (none vs. a little/some vs. a lot/unable)	Age, gender, education, race, Physical Activity Score for the Elderly, BMI, MMSE, lung and heart disease, diabetes mellitus, and baseline IADL difficulty, pain distribution, pain severity	Poisson regression	At 18 months follow-up, individuals with two (RR: 2.59, 95% CI: 1.10–6.09) or three (RR: 2.69, 95% CI: 1.34–7.79) persistent pain qualities had higher risk than those with one.
Makris UE. [48]	3-item scale (able vs. unable to complete task)	Age, female sex, non-white race, living alone status, less than high school education, depressive symptoms, overweight, physical frailty, cognitive impairment, ≥2 chronic conditions, hip weakness	Cox proportional hazards model	At 144 months follow-up, participants with restrictive back pain had higher hazard of disability (aHR: 2.33, 95% CI: 2.08–2.61).
Ćwirlej-Sozańska Ab. [49]	Lawton’s index (without limitation vs. ≥1 hard limitation)	Age, education, fall, adaptation of the interior, using assistive devices, physical activity, number of diseases, QOL	Logistic regression	Each 1-point increase on the VAS was associated with 21% higher odds of disability (aOR: 1.21, 95% CI: 1.06–1.36).
Connolly D. [28]	Lawton’s index (no difficulty vs. difficulty)	Gender, age, marital status, medical insurance, living status, education, employment status, loneliness, social connectedness score, time spent sitting, BMI, the presence of a chronic condition, fall in past year, smoking status, vision, hearing, number of medications, quality of life, depression, worry levels, self-rated memory, cognition, self-rated health and physical activity	Logistic regression	No significant association was found for participants troubled by pain.
Liang En W. [50]	Lawton’s index (independent vs. deficient)	Age, sex, marital status, ethnicity, education, employment, income, diabetes, hypertension, hyperlipidemia, BMI, fall, visual impairment, hearing impairment, cognition, depression, comorbidity, living arrangements, household, social support	Logistic regression	Participants with chronic pain had lower odds of IADL independence (aOR: 0.42, 95% CI: 0.20–0.90).
Shega JW. [23]	6-item scale (score range: 0 to 6)	-	Wald test	Participants with moderate or greater pain had significantly higher IADL scores than those with none to mild pain. (*p* < 0.001).
Yağci N. [51]	Lawton’s index (score range: 0 = dependent to 24 = independent)	-	*t*-test	The spinal pain group had lower scores than lower extremity pain group (*p* < 0.05).
Eggermont LH. [52]	3-item scale (no difficulty vs. difficulty or inability)	Age, sex, race, education, BMI, cognitive function, comorbid conditions, level of physical activity, daily analgesic use, and number of psychotherapeutic medications	Poisson regression	At the 18 months follow-up: Multisite pain (aRR: 2.14, 95% CI: 1.37–3.34) or widespread pain (aRR: 2.69, 95% CI: 1.61–4.50) compared to those with no pain were associated with higher risk. 3rd quartile of pain severity (aRR: 1.89, 95% CI: 1.16–3.08) had higher risk vs. 1st quartile. 2nd (aRR: 2.19, 95% CI: 1.24–3.86), 3rd (aRR: 2.22, 95% CI: 1.43–3.45), and 4th (aRR: 2.56, 95% CI: 1.55–4.22) quartile had higher risk vs. 1st quartile. Persistent multisite pain (aRR: 2.72, 95% CI: 1.86–3.97) was associated with higher risk over time. Pain in the back (aRR: 2.51, 95% CI: 1.59–3.94), hand/wrist (aRR: 2.89, 95% CI: 1.84–4.54), hip (aRR: 2.94, 95% CI: 1.79–4.85), knee (aRR: 2.04, 95% CI: 1.23–3.37), and feet (aRR: 2.20, 95% CI: 1.27–3.82) was associated with higher risk.
Shega JW. [24]	OARS (no help needed vs. any help needed)	Age, gender, race, education, depressed mood, co-morbidity (summary count), self-rated health	Logistic regression	Participants with moderate or greater noncancer pain had higher odds of disability both among cognitively impaired participants (aOR: 1.74, 95% CI: 1.15–2.62) and cognitively intact participants (aOR: 1.40, 95% CI: 1.20–1.63).
Buchman AS. [53]	OARS (no help needed vs. help needed or inability)	Age, sex, education, BMI. physical activity, cognition, depressive symptoms, vascular diseases, vascular risk factors	Cox proportional hazards model	At 5.6-year follow-up, each additional painful sites was associated with increased hazard (aHR: 1.10, 95% CI: 1.01–1.20).
Shega JW. [54]	OARS summary score (range: 0 = no help needed to 10 = impairment)	Age, gender, education, depression, comorbidity index, cognitive impairment	Linear regression	Participants with moderate or greater noncancer pain had higher disability scores (β: 0.17, 95% CI: 0.07–0.26).
Weaver GD. [55]	10-item scale (scores range: 0 = no limitations to 10 = limitations in all activities)	Age, gender, education, marital status, comorbidity index, BMI, cognitive status, depressive symptoms, SPPB, frailty status	Negative Binomial regression	Participants with pain severity had greater IADL limitations (β: 0.23, *p* < 0.01).
Gureje O. [56]	7-item scale (no impairment vs. impairment)	Age, sex	Logistic regression	Participants with chronic pain had higher odds of IADL disability (aOR: 4.2, 95% CI: 2.81–6.42).
Miu DK. [25]	Lawton’s index (total score)	Depressive symptomatology, age, and sleep quality	Logistic regression	No significant association was found for pain.
Mossey JM. [57]	Lawton’s index (independent vs. help needed with 1–2 activities vs. >2 activities)	Age, gender, self-rated health, number of medical conditions	Logistic regression	No significant association was found for pain.
Barberger-Gateau P. [58]	Lawton’s index (not dependent vs. dependent)	Age, sex, education, place of residence, dyspnea, visual impairment, hearing impairment, MMS, depression	Logistic regression	No significant association was found for joint pain.

Abbreviations: IADL = Instrumental Activities of Daily Living; OARS = Older Americans Resources and Service; BMI = Body Mass Index; MMSE = Mini-Mental State Examination; QOL = Quality of Life; SPPB = Short Physical Performance Battery; aOR = adjusted Odds Ratio; CI = Confidence Interval; RR = Relative Risk; aHR = adjusted Hazard Ratio Note: “-” indicates data not reported in the original publication.

## Data Availability

Not applicable.

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
