# Peer review of "A Systematic Review of the Association Between Pain and Instrumental Activities of Daily Living Disability in Community-Dwelling Older Adults"

_geriatrics, 2025, doi:10.3390/geriatrics10050113_

Round 1
Reviewer 1 Report
Comments and Suggestions for Authors
Thank you for the opportunity to review an interesting manuscript. However, there are some issues that need to be clarified and considered.
Title: The topic is interesting, but it's important to note that Instrumental IADLs are more complex than ADLs. The term used may pose challenges for older adults, particularly those who are bedridden or have severe illnesses.
Introduction: There is limited information provided regarding the impact of pain on IADLs. Please be recommended to include more details to strengthen the rationale for this study.
Methods: Based on a single database, this can lead to limited information. It is important to consider adding other sources. Additionally, older patients who are bed-ridden, at the end of life, or suffering from severe illness often have restricted to perform IADL due to their health conditions. This limitation may result in a misleading association between pain and IADL. Please take the eligibility criteria into consideration.
Results:
- In Table 1, the studies referenced as #[32], #[45], and #[52] indicate a mean age of ≥ 60 years. It would be helpful to specify what this mean age entails since participants might range from 60 to 64 years. Moreover, the eligibility criterion is for individuals aged 65 years and older; please consider this point.
- In Figure 1, the exclusion of full-text articles should align with the PICO framework, as the authors did not mention the basis for the exclusion criteria.
- The selected articles appear to be heterogeneous due to varying inclusion and exclusion criteria. Considering a subgroup analysis may enhance the findings.
Reviewer 2 Report
Comments and Suggestions for Authors
I congratulate you on your work. Below are some suggestions for improving your research
Line 71: I would recommend not limiting your search to PubMed alone. There are other sources, such as WoS and PEDro, among others, that could help strengthen your review. I also recommend updating your search since it's been more than six months since you performed it.
Line 80: I don't understand why the search was limited to English. Nowadays, there are multiple websites and some AIs that allow you to overcome the language barrier. I think this may be an important bias to consider today.
Line 90: What percentage of consistency in the search did both authors present, both when identifying and extracting the data? In this regard, I recommend using the Kappa coefficient (k) for this.
Line 93: I recommend conducting a quality assessment of the articles to give greater weight to the research. Expressing results based on articles of low methodological quality can lead to considerable bias in the research.
Line 95: Please provide the data synthesis of the study, and at a methodological level, what you were going to extract from the articles to indicate whether there was a relationship or not.
Line 110: Be careful with spaces, e.g., "…, 2021[35,36], …”
Line 119: Perhaps "pain site" isn't the most correct term; perhaps “pain location”...
Lines 131 to 194: Considering that the objective of their study is to verify the relationship between pain and IADL, I believe these sections take some of the weight off the analysis that appears later. Keep in mind that they are repeating data already shown in the tables. I encourage you to summarize these sections somewhat, highlighting only the most important aspects.
Line 320: The number of articles is not a conclusion. Please rewrite the conclusions.
Round 2
Reviewer 1 Report
Comments and Suggestions for Authors
Thank you to the authors for carefully responding to the comments. There are no more comments.
Reviewer 2 Report
Comments and Suggestions for Authors
No suggestions.